# Position: The Text-Centric Bias in Foundation Models Must Be Revisited for a Speech-First Future

**Deepak Babu Piskala** [1]

## Abstract

**This position paper argues that the machine learning community should prioritize speech-native architectures that treat audio as a first-class modality, anticipating the inevitable shift from text-dominated to speech-first data distributions.** Text dominates human-computer interaction not because it is cognitively natural, but because decades of interface design conditioned users to express knowledge through keyboards and search boxes. Recent advances in speech recognition and multimodal foundation models have removed the technical barriers to voice-based interaction; what remains is primarily a habit problem. As voice becomes habitual, the data ecosystem underlying machine learning will shift toward speech-native knowledge—with profound implications for model architecture, training efficiency, and evaluation paradigms. This paper examines the technical readiness of speech systems, identifies habit inertia as the primary adoption barrier, addresses alternative views that favor text-centric approaches, and outlines a research agenda for ML systems that anticipate speech-first data distributions.

## 1. Introduction

Modern AI systems overwhelmingly assume text as the primary medium of interaction, retrieval, and learning. Users type queries into search boxes, provide instructions through text prompts, and receive responses as rendered text. This assumption is rarely questioned—not because text is inherently optimal for human communication, but because decades of human-computer interaction have conditioned users to express intent through keyboards, forms, and graph-

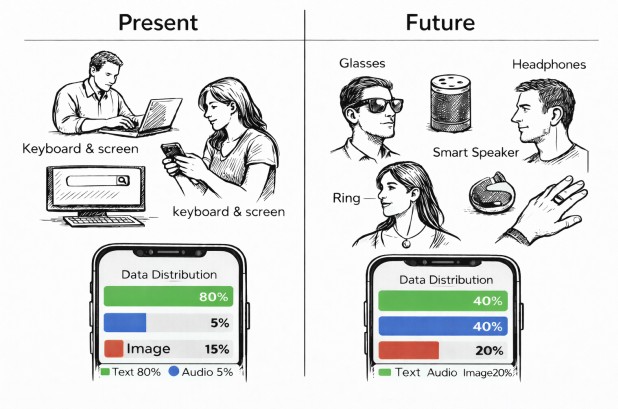

*Figure 1.* Evolution of human–computer interaction interfaces and their implications for training data distributions. As interaction shifts from keyboard- and screen-based inputs toward speech-first, wearable, and ambient devices, the dominant data modality transitions from text-heavy corpora to audio-rich representations.

ical interfaces. Text feels natural largely because it is habitual.

Speech, by contrast, has historically been treated as a convenience feature or accessibility tool. Yet spoken language is humanity's most fundamental communication modality, preceding writing by millennia and requiring no formal training to acquire. A child learns to speak years before learning to write; adults converse fluently while many struggle with written composition. The primacy of text in computing reflects infrastructure constraints and interface design choices, not cognitive naturalness.

Recent advances in machine learning have dramatically improved speech recognition and synthesis, bringing voice-based interfaces to technical parity with text in many contexts (Radford et al., 2022; OpenAI, 2024a). Modern speech recognition achieves near-human accuracy on standard benchmarks (Xiong et al., 2017), and neural text-to-speech systems produce indistinguishable-from-human audio (Shen et al., 2018). Speech-native foundation models like GPT-4o process audio end-to-end without intermediate text transcription, achieving response latencies within the range of human conversational turn-taking (OpenAI, 2024a). What remains is not primarily a modeling problem,

[1]Seattle, WA, USA. Correspondence to: Deepak Babu Piskala <prdeepak.babu@gmail.com>.

*Proceedings of the 43rd International Conference on Machine Learning*, Seoul, South Korea. PMLR 306, 2026. Copyright 2026 by the author(s).

but a habit problem: shifting both users and systems away from deeply ingrained text-first workflows. This transition is already underway: global voice assistant devices doubled from 4.2 billion in 2020 to 8.4 billion in 2024 (Statista Research Department, 2024).

Recent advances are enabling AI-native voice interfaces to migrate beyond keyboards and handheld screens into new form factors, including glasses, earbuds, rings, pendants, and ambient speakers. Unlike traditional voice assistants constrained to explicit wake words and discrete commands, these interfaces support continuous, low-friction interaction embedded in daily activity. As interaction moves from episodic typing to ambient speech, the manner in which users externalize intent and knowledge is likely to shift accordingly, with direct implications for the data distributions underlying future foundation models.

**Our position is this: The machine learning community should prioritize building speech-native foundation models—architectures that learn from audio as a first-class modality rather than grafting speech onto text-pretrained systems—because the data ecosystem underlying AI is poised to shift toward speech-first knowledge generation, and models designed for text-centric assumptions will be misaligned with future training distributions.**

### 1.1. Why This Matters for Machine Learning

This transition has profound implications for the future direction of machine learning research. Interfaces determine what data is created, stored, and reused for training. The dominance of text in AI systems reflects not just what humans *can* express, but what they have been *conditioned* to express through decades of keyboard-centric interaction. As voice becomes a primary interface—lower in effort, faster in expression, and more natural in form—knowledge will increasingly be generated in audio and video form, altering the statistical structure of future training data.

Consider the feedback loop:

1. **Interface shapes expression**: Search boxes condition users to express queries as keyword lists; voice enables natural language questions with context and nuance.

2. **Expression shapes data**: What users externalize becomes the training corpus for future models.

3. **Data shapes models**: The modalities, structures, and biases in training data determine what AI systems learn and how they represent knowledge.

This loop has favored text for decades. As it shifts toward speech, the implications cascade through the ML pipeline: from data collection and annotation, to model architecture

and training efficiency, to evaluation paradigms and deployment constraints. Figure 2 illustrates one dimension of this shift: as query complexity increases, speech becomes increasingly efficient relative to typing.

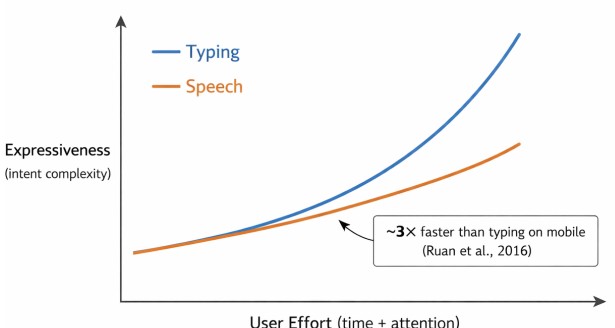

*Figure 2.* Comparison of input modalities: as query complexity increases, speech becomes increasingly efficient relative to typing. Speech input on mobile devices allows approximately 153 WPM, roughly three times the rate of smartphone typing (∼52 WPM) (Ruan et al., 2016).

### 1.2. Current Assumptions in Multimodal AI

Current multimodal architectures successfully integrate speech, vision, and text—but they do so with implicit assumptions about modality roles that may not hold as the data distribution shifts. Specifically:

- **Text is treated as the canonical representation**, with audio and video serving as auxiliary signals that are often collapsed back to text for reasoning and retrieval.

- **Modalities are treated symmetrically in training**, despite asymmetric costs in computation (continuous vs. discrete), annotation (transcription vs. labeling), and information density (compressed vs. redundant).

- **Evaluation focuses on text-centric tasks**, measuring how well models convert speech to text rather than how well they reason over audio-native knowledge.

As speech becomes habitual, these assumptions will misalign with the data ecosystem. ML research should anticipate this transition—not by abandoning text, but by recognizing that future intelligence will be trained on data that is spoken first, not written first.

### 1.3. Roadmap

The remainder of this paper proceeds as follows. Section 2 reviews advances in speech and language modeling that have made voice technically viable as a scalable data source.

Section 3 examines current multimodal architectures, introducing the concept of modality asymmetry and arguing that audio's role remains auxiliary despite its presence. Section 4 identifies habit inertia as the primary barrier to voice adoption, supported by illustrative user survey data. Section 5 presents and addresses alternative views that challenge our position. Section 6 outlines a call to action for the ML community. Section 7 concludes with implications for the field.

**Conflict of Interest Disclosure**

The author is employed as a Principal Researcher at Microsoft. This work was carried out independently of the author's employment and does not evaluate, benchmark, or promote any Microsoft product. No external funding or sponsorship was received for this research.

## 2. Advances in Speech and Language Modeling

For speech to function as a primary data source, recognition must be accurate and efficient at scale, and processing costs must be tractable. Recent advances have largely satisfied these prerequisites (see Section B for detailed historical context).

The trajectory of speech technology follows a clear arc: from brittle, speaker-dependent systems of the 1990s requiring hours of enrollment, through the deep learning revolution of the 2010s that replaced hand-engineered features with learned representations, to today's foundation models trained on hundreds of thousands of hours of multilingual audio. Each transition removed a critical barrier. Statistical HMMs gave way to neural acoustic models that captured long-range dependencies. Supervised pipelines yielded to self-supervised learning that eliminated the annotation bottleneck. Language-specific systems evolved into multilingual models with zero-shot cross-lingual transfer. The result is that speech recognition has transformed from a research curiosity to an infrastructure primitive—accurate enough for real-world deployment, efficient enough for on-device execution, and scalable enough to process the world's audio.

Speech carries information beyond words—emotion, intent, accent, and environment (Jurafsky & Martin, 2023). Modern ASR achieves near-human accuracy on standard benchmarks, with Word Error Rates (WER) below 5% in controlled settings. Speaking is approximately 3× faster than typing on mobile devices (Ruan et al., 2016), and on-device models now run locally on smartphones. Figure 3 illustrates the evolution toward modern self-supervised multimodal architectures.

**Modern Architectures.** Conformer (Gulati et al., 2020) combines convolution and self-attention for both local acous-

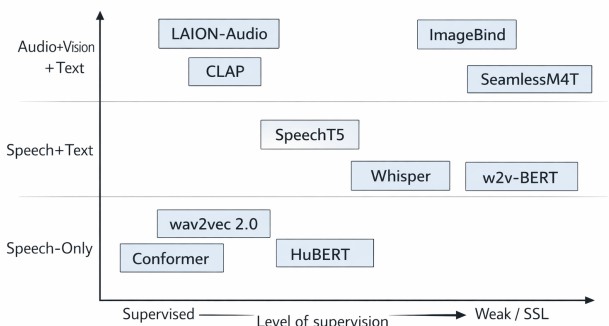

*Figure 3.* Evolution of speech and multimodal models, from supervised speech-only systems to self-supervised multimodal architectures that jointly process audio, text, and vision (Baevski et al., 2020; Hsu et al., 2021; Girdhar et al., 2023).

tic patterns and global context. Streaming transducers (Graves, 2012; Zhang et al., 2020) enable real-time recognition. OpenAI's Whisper (Radford et al., 2022), trained on 680,000 hours of weakly supervised multilingual data, achieves robust recognition across languages and accents without task-specific fine-tuning.

**Self-Supervised Learning.** Models like wav2vec 2.0 (Baevski et al., 2020) and HuBERT (Hsu et al., 2021) learn representations by predicting masked audio portions, analogous to masked language modeling. XLS-R (Babu et al., 2022) and USM (Zhang et al., 2023b) scale this to over 100 languages, reducing reliance on expensive labeled data.

**Speech Synthesis.** Neural TTS systems like WaveNet (van den Oord et al., 2016), Tacotron 2 (Shen et al., 2018), and FastSpeech 2 (Ren et al., 2020) produce natural-sounding speech. VALL-E (Wang et al., 2023) and Voicebox (Le et al., 2023) enable zero-shot voice cloning and multilingual synthesis, allowing voice interfaces to respond with fluid, human-like speech.

## 3. Multimodal Models and Audio's Role in AI

Despite the rapid progress described above, current multimodal architectures exhibit a fundamental asymmetry: *text remains the canonical representation, with audio grafted onto text-trained systems rather than learned natively.* Nearly all speech-language models initialize from text-trained LLM backbones—AudioPaLM uses PaLM-2, SpeechGPT uses LLaMA, Qwen-Audio builds on Qwen. This design choice has practical benefits (leveraging pretrained knowledge) but also consequences: these models inherit text-centric representations, reasoning patterns, and biases. Audio is discretized into tokens that approximate text structure, potentially losing the paralinguistic information that makes speech rich.

Humans perceive and interact with the world through multi-

ple senses—we see, hear, and touch, often simultaneously. Likewise, the most powerful AI systems are beginning to integrate modalities beyond just text. While early AI and information retrieval systems converted every problem into text (e.g., describing an image in words to answer questions about it), this loses significant information. Multimodal models aim to process data in the form it's generated.

Speech carries rich paralinguistic information: tone, volume, and background sounds provide context that pure text lacks. A user's voice command might convey urgency or frustration by tone, which an intelligent assistant could recognize and factor into its response. The ambient audio can reveal whether the user is in a quiet home or a noisy street, suggesting different interaction strategies (Sarikaya, 2017).

Recent large-scale AI models treat audio as a first-class citizen. OpenAI's Whisper model (Radford et al., 2022) was trained on 680,000 hours of multilingual audio-transcription data, achieving robust speech recognition across many languages and accents. Meta's ImageBind (Girdhar et al., 2023) learns a joint embedding space for six modalities at once—including images, audio, and text—without explicit supervision linking them. The model can "bind" the sound of ocean waves with the image of a beach, or match animal sounds to animal pictures, learning cross-modal associations analogous to human perception.

CLAP (Elizalde et al., 2022; Wu et al., 2022) enables zero-shot audio classification through contrastive language-audio pretraining. AudioLM (Borsos et al., 2022) and Sound-Storm (Borsos et al., 2023) generate audio while preserving speaker identity and prosody through hierarchical language modeling. Speech-LLM integration models like AudioPaLM (Rubenstein et al., 2023), SpeechGPT (Zhang et al., 2023a), Qwen-Audio (Chu et al., 2023), and Moshi (Défossez et al., 2024) enable end-to-end speech understanding and generation within large language models.

Research suggests that models learning from audio + text + images may achieve higher accuracy with less data than unimodal models, because each modality provides complementary information. While text data on the internet is finite and arguably reaching saturation for training giant LLMs, there is an abundance of audio and visual data that could continue scaling AI training. This hints that "beyond words"—incorporating audio and other modalities—is a promising path for advancing AI capabilities.

The McGurk Effect (McGurk & MacDonald, 1976) illustrates audio-visual interplay in human perception. When video shows a person saying "ga-ga" but audio plays "ba-ba," many people perceive a third sound like "da-da"—the brain fuses conflicting cues into a different perception. This demonstrates that our interpretation of speech can be drastically altered by visual context. For voice-based HCI, this implies that purely voice-only interfaces might sometimes confuse users in ways that multimodal interfaces could alleviate. A voice assistant on a phone or smart glasses could display visual cues to disambiguate what it heard, combining voice's speed with visual clarity.

### 3.1. Audio-Language Model Architecture

A key architectural pattern has emerged for building multimodal audio-language models (Figure 4). These systems use a large language model (LLM) checkpoint as a foundational backbone, extended with custom modal tokens to learn joint representations across modalities. Since speech is a continuous signal unlike discrete text (words or subwords), audio must first be *discretized* into tokens using self-supervised models like HuBERT (Hsu et al., 2021) or wav2vec 2.0 (Baevski et al., 2020). This extends the token vocabulary from text-only to include audio representations.

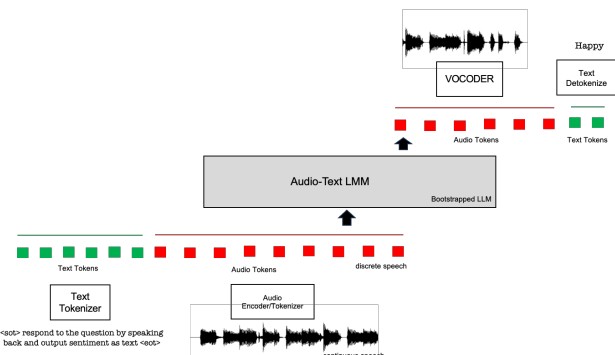

*Figure 4.* Audio-language model architecture. Input audio is tokenized via an audio encoder (e.g., HuBERT) and combined with text instructions. The LLM backbone autoregressively generates interleaved text and audio tokens, with audio tokens passed through a vocoder for speech synthesis.

The training pipeline typically involves: (1) **self-supervised pretraining** on joint text-audio tasks to align modalities in a shared representation space, and (2) **supervised instruction fine-tuning** (IFT) enabling diverse audio tasks. The model autoregressively generates interleaved text and audio tokens, with audio passed through a vocoder for synthesis.

Several recent models exemplify this pattern—AudioPaLM (Rubenstein et al., 2023) (PaLM-2 backbone), SpeechGPT (Zhang et al., 2023a) (LLaMA backbone with chain-of-modality prompting), and Qwen-Audio (Chu et al., 2023) (Whisper-based with 30+ audio tasks)—all demonstrating that text-trained LLMs can be extended to audio through vocabulary expansion and multi-stage fine-tuning.

### 3.2. Speech-Native Foundation Models in Production

A pivotal development in 2024–2025 has been the emergence of *speech-native* foundation models that process audio

end-to-end without intermediate text transcription. Unlike traditional cascaded pipelines (ASR → LLM → TTS), these models operate directly on audio tokens, preserving paralinguistic information and dramatically reducing latency.

OpenAI's GPT-4o represents a landmark in this evolution. The model processes audio natively, achieving response latencies as low as 232ms (average ∼320ms)—within the range of human conversational turn-taking (OpenAI, 2024a). This is a dramatic improvement over previous voice assistants that typically exceeded 1,000ms. The OpenAI Real-time API further enables developers to build low-latency, speech-to-speech applications with natural interruption handling (OpenAI, 2024b).

Google's Gemini Live API offers similar capabilities, supporting bidirectional voice and video streaming with "barge-in" functionality that allows users to interrupt model responses mid-sentence (Google Cloud, 2025a). Google frames this as a fundamental architectural shift: "moving away from rigid, multi-stage voice systems toward a single, real-time, emotionally aware, multimodal architecture" (Google Cloud, 2025b). These systems can detect emotional cues in the user's voice and adjust their responses accordingly.

These developments validate the speech-native architecture proposed in Figure 4. By processing audio directly rather than converting to text, these models preserve rich acoustic information—tone, emphasis, hesitation—that enables more natural and contextually appropriate responses. The commercial success of these systems demonstrates that speech-first interaction is not merely technically feasible but increasingly preferred by users for extended conversations.

**The Asymmetry Problem.** To summarize: the field has made speech *accessible* to LLMs, but has not yet made LLMs *native* to speech. Most multimodal models inherit text-centric architectures, with audio adapted to fit discrete token frameworks designed for language. GPT-4o and Gemini Live represent exceptions—truly speech-native systems— but they remain proprietary and rare. As speech data grows in volume and importance, architectures that do not inherit text biases may perform better on audio-native tasks. This asymmetry has implications for evaluation (current benchmarks favor text-centric tasks), training efficiency (audio requires different compute/data tradeoffs), and model design (preserving paralinguistic information requires architectural innovation).

## 4. Challenges in Voice Interface Adoption

If the technical barriers to voice have largely been removed (Section 2) and multimodal models can process speech (Section 3), why hasn't voice replaced text as the dominant interface? We argue habit inertia is now an underappreci-

ated residual barrier: once technical limitations cross usability thresholds, the accumulated behavioral and institutional infrastructure built around text becomes the dominant remaining friction. Crucially, this habit is itself a *product* of prior technical constraints. Users were trained on keyboards because that was what worked. Today, as accuracy and latency barriers fall, what persists is the feedback loop those barriers created: habits shaped the data, data shaped the models, and models now reinforce the text-centric default.

Despite rapid technological improvements, several key challenges have prevented voice interfaces from completely supplanting traditional input methods (Clark et al., 2019; Deshmukh & Sajja, 2024). As summarized in Figure 5, it's not just about achieving low error rates in the lab; it's about creating a user experience compelling enough to change deeply ingrained habits (Dutsinma & Temdee, 2022).

*Figure 5.* Taxonomy of barriers to voice adoption: technical challenges (accuracy, latency, noise robustness), social factors (public awkwardness, privacy concerns), and behavioral factors (habit inertia, limited discoverability) (Klein et al., 2024).

**Accuracy and Reliability.** While ASR systems are much improved, even a 5% error rate can significantly impact user trust if those errors occur at critical moments (Kim et al., 2021). Misrecognizing a single word ("add" vs "delete") can lead to undesirable outcomes. Users notice and remember when a voice assistant makes a mistake, discouraging use for important tasks (Lahoual & Frejus, 2019). Accuracy varies with accent, dialect, or language—many systems perform best on American English and see higher error rates for other accents, leading to inequities in user experience (Xiong et al., 2017). In professional domains such as medicine, law, and finance, this challenge is acute: recent benchmarking shows that even state-of-the-art models including Whisper and audio-language models like Qwen-Omni exhibit a "context-utilization gap"—failing to leverage available contextual information to improve entity-level accuracy in high-stakes settings (Piskala, 2025).

**Latency (Response Time).** Conversation is interactive and humans are highly sensitive to timing. Studies on conversational systems suggest an ideal response latency of only a few hundred milliseconds (perhaps 200–500ms after

the user finishes speaking) to mimic natural dialogue flow (Stivers et al., 2009; Levinson & Torreira, 2015; Levinson, 2016). However, as shown in Figure 6, today's voice assistants often have latencies exceeding one second or more due to endpointing (Liang et al., 2022; Ding et al., 2020), network transmission, inference, and synthesis (Meyer et al., 2023). Figure 7 further illustrates the gap between human conversational timing and current voice assistant response times.

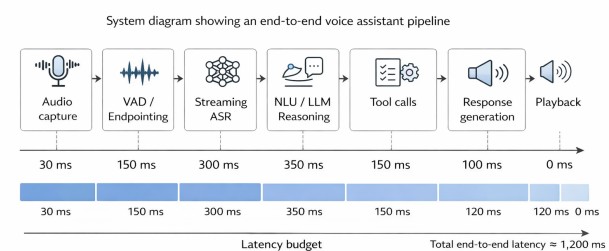

*Figure 6.* End-to-end voice assistant pipeline showing latency budget. Each stage—audio capture, VAD/endpointing, streaming ASR, NLU/LLM reasoning, tool calls, response generation, and playback—contributes to total latency of ≈1,200ms, far exceeding the 200ms human conversational norm.

**Noise and Environmental Factors.** Unlike typing, speaking is highly affected by environment. In noisy settings (crowded streets, cafes, moving cars), even humans struggle to understand speech—the classic "cocktail party problem" (Li et al., 2014). Background noise or other speakers confuse ASR systems, leading to frequent errors. Data augmentation with reverberant speech has helped improve robustness (Ko et al., 2017; Park et al., 2019). Using voice in public or crowded places is also socially impractical—many users feel uncomfortable talking to devices in front of others (PwC Consumer Intelligence Series, 2018; Moorthy & Vu, 2014).

**User Habits and Intuitiveness.** Decades of computing with keyboards and touchscreens have shaped user habits (Porcheron et al., 2018). For many people, text-based input feels "natural" simply because it's been the default for so long. Some users wouldn't naturally think to use voice—either forgetting the capability exists or being uncertain what assistants can handle (Cowan et al., 2017). Many cite "limited knowledge of the full breadth of capabilities" as a reason they stick to basic commands (Lahoual & Frejus, 2019).

**Privacy and Trust.** Voice assistants have faced privacy concerns since their inception (Lau et al., 2018; Cho et al., 2019). Reports of devices "always listening" and human reviewers accessing recordings have made people wary (Meng et al., 2021). Around 18% of consumers familiar with voice technology have never used a voice assistant, with privacy cited as a primary reason (PwC Consumer Intelligence Se-

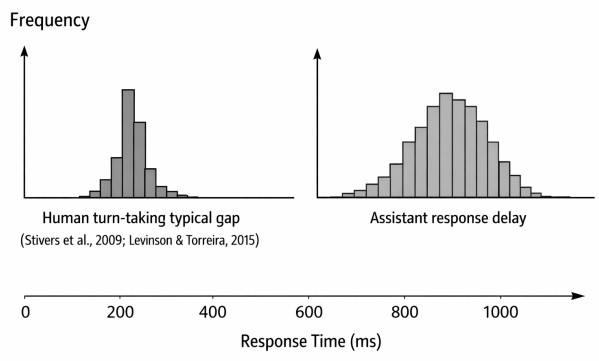

*Figure 7.* Turn-taking timing comparison: human conversational gaps center around 200ms (Stivers et al., 2009), while current voice assistants typically exceed 1,000ms.

ries, 2018). Unlike text which leaves a clear log, voice interactions feel ephemeral—users may not know what was recorded or stored.

**Dialog Limitations.** Current voice assistants are mostly geared toward single-turn commands (Ram et al., 2018; Hoy, 2018). They falter with multi-turn conversations requiring context memory (Henderson et al., 2014). If you ask "Find Italian restaurants nearby" then follow up with "Book a table at the first one," many systems struggle to link those utterances. This inability to hold context makes interactions feel unnatural compared to human conversation.

### 4.1. Empirical Evidence from User Survey

To ground these challenges empirically, we conducted an informal survey (N=200) of voice interface users.[1] As shown in Figure 8, while 76% of respondents use voice interfaces at least occasionally, significant barriers persist: speed concerns (48%), quality issues (43%), single-turn limitations (38%), social discomfort (33%), privacy concerns (29%), and lack of intuitiveness (19%).

The pattern is revealing: the top barriers, perceived slowness and quality concerns, reflect latency and accuracy issues that are rapidly improving. Yet behavioral factors persist: 100% of voice users reported using voice at home, but only 25% use it at work and even fewer in public spaces. Among those who rated recognition quality, only 6% described it as "Great" while 69% rated it "Average." These findings underscore that technical improvements alone are insufficient; voice interfaces must also address user perception, social context, and discoverability to achieve mainstream adoption.

---

[1]This survey is presented as illustrative evidence of barrier patterns, not as statistically representative research.

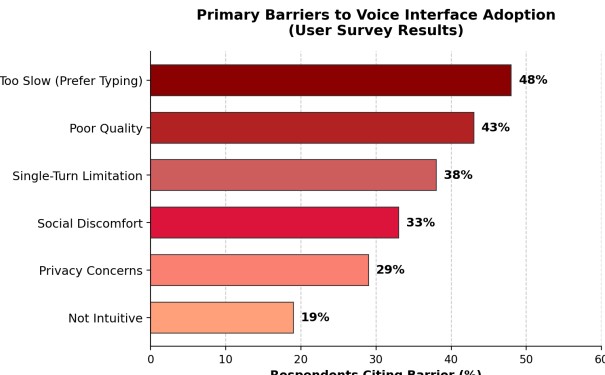

*Figure 8.* User survey results on barriers to voice interface adoption. Speed concerns and quality issues are the most frequently cited barriers, followed by single-turn limitations and social discomfort in public contexts.

**Survey limitations.** The survey was conducted as an informal online poll and is subject to self-selection bias toward technology-literate and younger respondents. We do not have controlled demographic or recruitment data. We therefore present these results as illustrative evidence of barrier patterns rather than as a representative population study. Even under these caveats, the barrier ranking (speed, quality, single-turn limitations, social discomfort, privacy) and the sharp home-versus-public usage gap (100% vs. 25%) are informative signals about which barriers dominate, even if absolute magnitudes are uncertain.

**Implications for ML Research.** Habit inertia is not merely an HCI problem—it has direct consequences for machine learning. As long as users default to text, the data ecosystem remains text-dominated, reinforcing text-centric model development. Breaking this cycle requires either (1) interfaces compelling enough to change habits, or (2) models capable of learning from the speech data that *does* exist (podcasts, meetings, lectures, voice messages) even if it's not explicitly curated for training.

## 5. Alternative Views

Our position—that the ML community should prioritize speech-native foundation models—faces several credible counterarguments. We present and address the strongest objections here.

### 5.1. "Text is More Computationally Efficient"

**The objection**: Audio processing is 10–100× more expensive than text processing for equivalent semantic content (Section A). Given finite compute budgets, investing in speech-native models diverts resources from text-based systems that can train on more data per FLOP. The scaling laws that drive LLM progress favor data-efficient modalities.

**Our response**: This argument assumes the current cost ratio is fixed. Self-supervised learning has already reduced audio processing costs dramatically—wav2vec 2.0 and HuBERT enable training without transcription labels. Hierarchical tokenization (AudioLM, SoundStorm) compresses audio to 50–75 tokens/second, approaching text density. More fundamentally, this objection optimizes for current data distributions. If future knowledge is increasingly generated as speech (podcasts, meetings, voice messages), text-only models will be trained on an increasingly biased sample of human knowledge—those portions that happened to be transcribed or written. The question is not which modality is cheaper today, but which will be more representative tomorrow.

### 5.2. "Paralinguistic Information Doesn't Justify the Cost"

**The objection**: The additional information in audio (tone, emotion, speaker identity) may not justify the computational overhead for most ML tasks. Text captures the semantic content that matters for reasoning, retrieval, and generation. Paralinguistic features are noise, not signal.

**Our response**: This objection reflects a text-centric definition of "what matters." For many real-world applications—customer service, healthcare, education, mental health support—emotional state and speaker intent are *central* to the task, not peripheral. A tutoring system that cannot detect student frustration, or a customer service agent that cannot recognize an angry caller, is fundamentally limited. Moreover, paralinguistic information provides disambiguation that text lacks: sarcasm, rhetorical questions, and emphasis all change meaning without changing words. Current text-only models struggle with these distinctions because they were trained to do so—not because text is inherently sufficient.

### 5.3. "User Preference for Text Reflects Genuine Advantages"

**The objection**: Users prefer text not merely from habit, but because text has genuine advantages: it's private (no one overhears), precise (no recognition errors), scannable (can review before sending), and archivable (searchable logs). These advantages will persist even as speech technology improves.

**Our response**: We acknowledge these advantages are real, and want to carve out where text-first remains the right default rather than an artifact of habit:

- **Search and retrieval**: discrete tokens index efficiently, and even spoken queries benefit from text-shaped intermediate representations.

- **Non-linear editing**: revising a document, refactoring code, or rearranging an outline requires random-access manipulation that speech does not naturally support.

- **Formal documentation**: contracts, specifications, and academic writing demand precision and immutability that speech production does not afford.

- **Silent and asynchronous contexts**: messaging in shared spaces, archival reading, and review workflows.

- **Accessibility for deaf and hard-of-hearing users**: text is the primary modality, and speech-native systems must complement rather than replace text pipelines.

Our claim is not that speech displaces text. It is that the field's architectural defaults (text-pretrained backbones, text-centric benchmarks, text-shaped tokenizers) implicitly treat speech as the edge case. Both modalities deserve first-class treatment, with each preferred where it genuinely wins.

### 5.4. "The Habit Shift May Never Happen"

**The objection**: Predicting behavioral shifts is notoriously difficult. QWERTY persisted despite Dvorak; PDAs and tablets coexisted for decades before smartphones dominated. Users may never adopt voice at scale, rendering speech-native models a solution to a nonexistent problem.

**Our response**: The comparison to QWERTY is instructive: QWERTY persisted because the switching cost exceeded the benefit for most users. We argue the switching cost for speech is lower (no new skill to learn) and the benefit higher ($3\times$ input speed, hands-free operation, accessibility). More importantly, the shift is already visible in specific domains: voice search queries now exceed 20% of mobile searches; smart speaker adoption continues to grow; and voice message usage is rapidly increasing in messaging apps worldwide. The question is not *whether* speech usage will grow, but *how fast*—and whether the ML community will be prepared.

## 6. Call to Action

The preceding analysis motivates specific actions for different stakeholders in the ML ecosystem. We propose concrete steps toward a speech-native future.

### 6.1. For Researchers: Three Concrete Directions

We highlight three research directions whose outcomes would distinguish speech-native from text-pretrained approaches in measurable ways, making them suitable as hypothesis-driven targets rather than open exhortations.

1. **Audio-first pretraining**: Train foundation models on audio from the start, rather than fine-tuning text-pretrained backbones on audio. Open questions: Do audio-first models develop different internal phonological structure? Are they more robust to non-standard accents and code-switching? Do they exhibit different scaling laws than text-then-audio models at matched compute?

2. **Semantic audio tokenization**: Current tokenizers (HuBERT, wav2vec 2.0) cluster on phonetic or acoustic similarity. Develop tokenizers whose latent spaces reflect semantic and paralinguistic structure hierarchically, preserving content, prosody, and speaker characteristics at distinct levels usable by downstream LLMs without forcing collapse to text.

3. **Evaluation without text intermediation**: Build benchmarks that measure reasoning *over* audio directly, including comprehension, summarization, multi-turn dialogue, and paralinguistic inference, without ASR-then-NLU pipelines. ProfASR-Bench (Piskala, 2025) provides a template for domain-specific, context-aware audio evaluation; broader audio-native counterparts to MMLU and HumanEval are missing.

### 6.2. Privacy as a First-Order Architectural Constraint

Privacy is not a deployment afterthought for speech-native systems. It is a first-order architectural constraint that shapes data pipelines, model design, and training infrastructure. Voiceprints, ambient sounds, and continuous capture create risks that text systems do not face. Any credible speech-native research agenda must therefore treat the following as core, not peripheral:

- **Consent-aware data pipelines** that distinguish opted-in content (podcasts, deliberate recordings) from ambient capture, with provenance tracked through training.

- **On-device processing** for sensitive audio, with cloud round-trips only for content the user has explicitly released.

- **Federated learning adapted for speech**, accommodating non-IID acoustic distributions across devices and speakers.

- **Differential privacy for continuous audio signals**, where standard text-DP guarantees do not transfer cleanly.

The GPT-4o system card already treats speaker identification and voice generation as model-specific safety risks (OpenAI, 2024a); the research community should formalize this rather than treat it as proprietary practice.

## 6.3. Multilingual Robustness and Accent Equity as Core Blockers

ASR accuracy still drops substantially for low-resource languages and non-standard accents (Xiong et al., 2017), and these populations are often the most speech-first in daily life: communities with limited literacy infrastructure, oral knowledge traditions, or dialectal variation underserved by standard ASR. The speech-native argument is therefore both more urgent and harder to realize there. We treat multilingual robustness and accent equity not as an open problem for future work, but as a core research blocker that any speech-native pretraining agenda must address from the start, alongside scale, efficiency, and privacy, rather than after architectures stabilize on English.

## 6.4. For Industry Practitioners

1. **Invest in speech data infrastructure**: The podcasts, meetings, and voice messages being generated today are tomorrow's training data. Building pipelines to collect, clean, and curate speech data, with appropriate consent and privacy protections, is a strategic investment.

2. **Deploy speech-native interfaces**: Rather than retrofitting voice onto text-designed products, design experiences where speech is the primary modality. This generates training signal for speech-native models and shifts user habits.

3. **Open-source speech-native models**: GPT-4o and Gemini Live are proprietary. The research community needs open alternatives to study speech-native architectures, identify failure modes, and iterate on designs.

## 6.5. For the Community

1. **Develop audio-native benchmarks**: Current evaluation suites (GLUE, SuperGLUE, MMLU) are text-centric. Audio-native benchmarks complement Section 6.1 by providing shared targets for community progress on comprehension over audio rather than transcription accuracy alone.

2. **Engage with HCI researchers**: The habit barriers we identified (Section 4) require interdisciplinary solutions. ML researchers should collaborate with HCI, cognitive science, and design communities to build interfaces that users actually want to speak to.

## 7. Conclusion

This position paper has argued that the ML community should prioritize speech-native foundation models that treat audio as a first-class modality. The technical barriers have largely been removed—modern ASR achieves near-human accuracy, speech-native models demonstrate sub-300ms latency, and self-supervised learning has made audio processing tractable at scale. What remains is primarily habit inertia: users default to text not because it is superior, but because decades of keyboard-centric interfaces have made it familiar. Meanwhile, current multimodal architectures exhibit text-centric asymmetry, grafting audio onto text-pretrained backbones rather than learning it natively.

As voice becomes habitual, the data ecosystem underlying ML will shift toward speech-first knowledge generation, and models designed for text-centric assumptions will be increasingly misaligned with future training distributions. The transition will not happen overnight, and text will remain important—but the ML community should anticipate this shift rather than react to it. Models that can learn from the speech data that already exists (podcasts, meetings, lectures, voice messages) and architectures that preserve paralinguistic information will be better positioned for a future where humans increasingly speak their knowledge before writing it down.

## Acknowledgments

The authors acknowledge minimal use of generative AI writing assistants for grammar correction and stylistic refinement. All conceptual development, technical analysis, and intellectual content are solely the work of the authors.

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

# A. Compute Cost Derivation: Audio vs. Text

This appendix derives the 10–100× compute cost ratio between audio and text processing claimed in Section 5. The estimate depends on representation choice and task, but the order of magnitude is robust across reasonable assumptions.

## A.1. Token Count Comparison

Consider representing the same semantic content—a 2-second utterance saying "The cat sat on the mat"—across modalities:

| Representation | Tokens | Ratio to Text |
|---|---|---|
| Text (BPE) | ∼7 | 1× |
| Audio (HuBERT @ 50/sec) | ∼100 | 14× |
| Audio (SoundStream @ 75/sec) | ∼150 | 21× |
| Raw waveform (16kHz) | 32,000 | 4,571× |

*Table 1.* Token counts for equivalent 2-second semantic content.

**Sources:** HuBERT produces approximately 50 tokens/second using k-means clustering over learned representations (Hsu et al., 2021). SoundStream and EnCodec use residual vector quantization at 50–75 tokens/second (Zeghidour et al., 2021; Défossez et al., 2022). Standard audio sampling is 16kHz for speech.

## A.2. Transformer Compute Scaling

For transformer architectures, self-attention complexity scales as $O(n^2)$ where $n$ is sequence length. For the same semantic content:

$$\text{Text attention ops} \approx 7^2 = 49 \tag{1}$$
$$\text{Audio attention ops (HuBERT)} \approx 100^2 = 10,000 \tag{2}$$
$$\text{Ratio} \approx 204\times \tag{3}$$

This ∼200× ratio for attention operations dominates compute for large transformers. However, modern architectures use techniques that reduce this gap:

- **Linear attention** approximations (Performer, Linear Transformers) reduce complexity to $O(n)$, lowering the ratio to ∼14×.

- **Chunked/streaming processing** processes audio in fixed-size windows, amortizing cost.

- **Hierarchical tokenization** compresses audio to fewer tokens at higher levels.

## A.3. Empirical Estimates

**Whisper** (Radford et al., 2022): The Whisper-large model (1.5B parameters) processes 30 seconds of audio in approx-imately 3 seconds on an A100 GPU. An equivalent text query (∼75 words) would require <0.1 seconds, suggesting a ∼30× ratio for inference.

**Training data scale**: Whisper was trained on 680,000 hours of audio, which when transcribed yields approximately 10 billion words—comparable to early GPT-scale text corpora. However, processing this audio required substantially more compute than processing equivalent text tokens due to the higher sampling rate and longer sequences.

**GPT-4o** (OpenAI, 2024a): OpenAI reports that GPT-4o's audio processing uses a "native" audio encoder that avoids intermediate text transcription. While exact compute costs are not disclosed, the 232–320ms response latency for audio (vs. ∼100ms for text) suggests audio processing adds ∼2–3× latency even in optimized systems.

## A.4. Summary

The 10–100× range captures the spectrum of audio processing costs:

- **Lower bound (∼10×)**: Heavily compressed audio tokens (50/sec) with efficient architectures (linear attention, streaming).

- **Upper bound (∼100–200×)**: Standard transformer self-attention over longer audio sequences.

- **Raw waveforms**: >1000× more expensive, rarely used directly in modern systems.

This cost asymmetry motivates our research agenda on compute-efficient audio processing (Section 6). As audio data grows in importance, reducing this gap through architectural innovation becomes increasingly critical.

# B. Historical Context: Speech Recognition Development

This appendix provides historical context for the advances discussed in Section 2.

## B.1. Statistical Era (1990s–2000s)

Early speech recognizers relied on statistical models: Hidden Markov Models (HMMs) for acoustic modeling and n-gram language models for predicting word sequences (Jurafsky & Martin, 2023). These systems achieved moderate accuracy in controlled settings but struggled with variability in speakers, noise, and domain (Li et al., 2014). Training required extensive labeled corpora and careful acoustic feature engineering (MFCCs, PLPs). Systems were typically speaker-dependent or required enrollment; far-field microphone performance was poor.

## B.2. Deep Learning Revolution (2012–2017)

Deep learning transformed speech recognition through two key innovations:

**Neural acoustic models**: Recurrent neural networks (RNNs), especially LSTMs, replaced GMM-HMM acoustic models, capturing long-range temporal dependencies (Chan et al., 2016; Chorowski et al., 2015). Convolutional neural networks provided translation-invariant feature extraction.

**End-to-end architectures**: Sequence-to-sequence models with attention (Listen, Attend and Spell (Chan et al., 2016)) and CTC-based approaches eliminated the need for separate phoneme alignment, simplifying training pipelines.

A milestone came in 2017 when Microsoft researchers achieved 5.1% WER on the Switchboard benchmark (Xiong et al., 2017)—roughly on par with human transcribers. This was accomplished using ensemble neural acoustic models (CNNs + LSTMs) combined with strong language models and GPU-accelerated training on massive datasets (Godfrey et al., 1992). However, this benchmark performance did not generalize: accuracy degraded substantially in noisy environments, with distant microphones, or for speakers with accents or underrepresented languages (Ko et al., 2017).

## B.3. Self-Supervised and Multilingual Era (2018–Present)

The most recent era has been characterized by self-supervised pretraining and massive multilingual systems:

**Self-supervised pretraining**: wav2vec 2.0 (Baevski et al., 2020) and HuBERT (Hsu et al., 2021) learn representations from unlabeled audio by predicting masked portions, dramatically reducing labeled data requirements. Fine-tuning with only 10 minutes of labeled data achieves competitive WER.

**Multilingual scaling**: XLS-R (Babu et al., 2022) extends self-supervised learning to 128 languages using 436K hours of speech. USM (Zhang et al., 2023b) further scales to 100+ languages with strong zero-shot cross-lingual transfer.

**Weak supervision at scale**: Whisper (Radford et al., 2022) demonstrated that training on 680,000 hours of weakly supervised (internet-sourced) multilingual data produces robust recognition across languages and accents without task-specific fine-tuning—a fundamentally different scaling paradigm from the labeled-data-intensive approaches of earlier eras.

## B.4. Implications

This historical progression illustrates how each era removed specific barriers: HMM limitations on context modeling, labeled data requirements, and language-specific engineering. The current era has achieved technical readiness for speech as a scalable data source—the remaining barriers are behavioral, not technical.

