# OpenReview forum: "Position: *Beyond Text* The Text-Centric Bias in Foundation Models Must Be Revisited for a Speech-First Future"
_ICML.cc/2026/Position_Paper_Track — ICML 2026 Position Paper Track spotlight_

### Official Review · Reviewer_FpQw · 2026-03-12

**Significance:** 3
**Argument Clarity:** 3
**Rating:** 5
**Confidence:** 4

**Questions:**

- Other than the issue of lacking of open-source off-the-shelf speech-native foundation models, what other notable examples of pushback against speech-native adoption in the field are there?
- How do the authors empirically justify that the primary barrier to voice interface adoption is habit inertia?
- Why argue only for speech native instead of making a general argument for being multimodal native?

**Alternative Views Section:**

Yes

**Compliance With Llm Reviewing Policy A Conservative:**

Affirmed.

**Discussion Potential:**

2

**Final Justification:**

Main writing-related concerns have been addressed. Reviews from other reviewers suggest that there is sufficient interest in this topic. I therefore raise my rating.

**Paper Summary:**

The paper proposes that future multi-modal LLMs move beyond the current paradigm where we start with a text-only base model and integrate speech capabilities afterwards. They argue that the distribution of data will shift towards large amounts of audio-only data. When that happens, choosing model hyperparameters based on text-centric tasks as we do now may misalign with the end goal of modeling audio/speech-native capabilities.

**Position:**

Yes

**Position In Title:**

Yes

**Related Work:**

4

**Strengths And Weaknesses:**

Strengths:
- The paper provides a detailed history of speech language modeling
- The paper presents reasonable supporting arguments, such as the issue of latency in a cascading pipeline involving ASR or TTS, the information lost in text-only representations compared to speech, and examples of recent observed benefits in the few available speech-native models.

Weaknesses:
- This work targets a narrow argument for being speech native, but the arguments generally hold for a broader push for being multi-modal native.
- Section 4 starts by stating that voice hasn't supplanted text as the dominant interface due to habitual reasons, but following supporting paragraphs list a number of technical reasons as well (ASR accuracy, latency, noise). I don't think the habitual inertia being the _primary_ barrier is explicitly supported by the user study in figure 8 either.
- In Alternative view 5.2, I have a hard time imagining someone knowledgeable claiming paralingustic features are noise, not signal. The objection seems valid, but perhaps there is a better example argument to use here?
- Alternative view 5.4 feels like a strawman argument to me. I don't think arguing that the switching cost from text to speech interfaces being lower than from QWERTY to Dvorak is particularly meaningful.
- I'm not very convinced that this is a topic that will likely inspire discussion because I don't think this is a controversial position. Examples include the recent release of multi-modal native omni models such as GPT-4o and Gemini (also noted in the paper), as well as vision-native models such as Kimi-k2.5. I think the potential for this position to spark conversation would have been much more significant a year ago.

**Support:**

3

---

> ### Author Rebuttal · Authors · 2026-03-29
>
> We thank the reviewer for the direct and substantive engagement. Before addressing specific points, we want to restate our thesis clearly: the ML community should stop treating speech as an add-on to text-pretrained systems, because a growing share of human knowledge is being created in spoken form and text-pretrained architectures discard signal that speech carries natively. The question is not whether text remains valuable - it obviously does - but whether text should remain the unchallenged default substrate for foundation model pretraining. We argue it should not.
>
> **W1 (narrowness / multimodal-native):** We disagree that the argument generalizes freely. Speech is not just another perceptual modality like vision - it is the primary modality through which humans *create and exchange knoweldge* in real time. Meetings, lectures, voice messages, podcasts, conversational search, clinical consultations all generate knowledge natively in spoken form. When routed through text-pretrained models, the text bottleneck imposes a representational structure (discrete tokens optimized for written language) never designed for this signal.
>
> Vision, by contrast, is primarily an *observation* modality - humans perceive visually but rarely create knowledge by generating images. Speech shapes *what knowledge exists and how it is structured*. This is why the interface -> expression -> data -> model feedback loop (Section 1.1) has specific force for speech that it does not have for other modalities. We will add one paragraph making this distinction explicit, but we do not concede this is a generic multimodal argument.
>
> **W2 (habit inertia):** Fair critique of the *phrasing*. We do not claim habit is the only barrier. The intended argument is: once technical barriers cross usability thresholds, habit becomes an underappreciated residual factor explaining why adoption lags capability. The survey illustrates barrier patterns; the footnote on page 6 already says so.
>
> However we want to push back on the suggestion that technical barriers alone explain the gap. 76% of respondents use voice at least occasionally, 100% at home, but only 25% at work or public. If purely technical, usage wouldn't vary this sharply by social context.
>
> Importantly, the technical and behavioral barriers are not independent. Decades of technical limitations in speech interfaces *produced* the text-centric habits. Users didn't choose text because they preferred it - they were trained on it because it was the only thing that worked. Now that technical barriers are falling, what remains is accumulated behavioral and institutional infrastructure built around text. This is why we call it inertia rather than preference - and why the feedback loop matters: habits shaped the data, data shaped the models, models now reinforce the text-centric default.
>
> **W3 (Alt View 5.2):** Fair point. We will replace with a stronger objection: "For most high-value downstream tasks - reasoning, retrieval, summarization - semantic content dominates, and paralinguistic signal does not justify the 10-100x computational overhead at pretraining scale." Our response: the set of tasks where paralinguistic signal is central (healthcare, education, customer service, safety-critical systems) is already large and growing.
>
> **W4 (Alt View 5.4):** Agreed, the QWERTY analogy is strained. The real point: the question is not whether a wholesale behavioral switch occurs, but whether marginal growth of speech-generated data creates architectural misalignment. Voice search exceeds 20% of mobile queries, voice messaging dominates in WhatsApp/WeChat, podcast and meeting content is growing exponentially. None of this requries a "full switch."
>
> **W5 (not controversial):** We respectfully and strongly disagree. GPT-4o and Gemini Live show two proprietary labs exploring speech-native architectures. The open research community has not followed. AudioPaLM, SpeechGPT, Qwen-Audio, Moshi - all initialize from text-pretrained backbones. Major benchmarks (GLUE, SuperGLUE, MMLU, HumanEval) are entirely text-centric. ICML, NeurIPS and ICLR remain overwhelmingly text-first. If this were uncontroversial the field would already be acting on it. The existance of GPT-4o means one closed system demonstrated feasibility - not that the community has embraced speech-native pretraining as a priority.
>
> **Promised revisions:** (1) Narrow habit inertia phrasing, clarify tech caused the habits. (2) Rewrite Alt Views 5.2 and 5.4. (3) Add paragraph on why speech specifically as a knowledge creation modality. (4) Expand text-superiority carve-outs and research directions. (5) Elevate privacy and multilingual equity to core constraints.

---

> > ### Author Rebuttal · Reviewer_FpQw · 2026-04-03
> >
> > Based on the feedback from other reviewers, it seems that this topic would likely be of general interest.  I think the overall writing is decent and the promises to fix the phrasing for habit intertia are sufficient for satisfying my concerns.
> >
> > Regarding images, I did want to point out that the current focus in native multimodal pretraining involving images is strongly focused on OCR and OCR-dependent downstream tasks such as interpreting figures, graphs, and graphical user interfaces. OCR can be considered the visual analog of ASR, so I don't think the differences between vision-native and speech-native setups are that clear cut from my point of view. That said, I think the interest as expressed by other reviewers on this topic is sufficient so I'm happy to raise my rating in response to the rebuttal.

---

### Official Review · Reviewer_8HwM · 2026-03-13

**Significance:** 4
**Argument Clarity:** 4
**Rating:** 5
**Confidence:** 4

**Questions:**

1. Could the authors elaborate on the path to resolving the compute cost asymmetry? While Appendix A mentions hierarchical tokenization and linear attention, how realistically do these scale for trillion-parameter models compared to established text architectures?

2. Regarding the user survey, can the authors provide more demographic context (e.g., age, geographic location) to better validate the $N=200$ sample and the "habit inertia" claims?

3. How do the authors propose handling the inherent unsearchability and privacy risks of archiving ambient speech data at the scale required for foundation model pretraining, beyond the brief mention of federated learning and on-device models?

**Alternative Views Section:**

Yes

**Compliance With Llm Reviewing Policy A Conservative:**

Affirmed.

**Discussion Potential:**

3

**Paper Summary:**

The paper argues that the machine learning community must transition from text-centric foundation models to natively speech-first architectures. The authors contend that the current dominance of text interfaces is largely due to habit inertia and historical constraints rather than cognitive naturalness. Because recent technical advancements have resolved major accuracy and latency barriers in voice technology, the authors predict an impending shift toward speech-first data generation. Consequently, they argue that simply grafting audio processing onto text-centric models is inadequate. A true transition to speech-native pretraining is required to capture rich paralinguistic information, align with future data ecosystems, and overcome the current modality asymmetry.

**Position:**

Yes

**Position In Title:**

Yes

**Related Work:**

3

**Strengths And Weaknesses:**

Strengths:

Timeliness and Relevance: The position is highly relevant given the recent commercial releases of speech-native models like GPT-4o and Gemini Live. The paper effectively captures a current inflection point in multimodal AI.

Clear Argumentation: The paper clearly articulates a logical cascade: interface shapes expression, expression shapes data, and data shapes models. This provides a strong foundation for why ML researchers should care about human-computer interaction habits.

Good Counter-Arguments: The "Alternative Views" section addresses the strongest objections, particularly the compute-cost asymmetry between text and audio processing.

Actionable Advice: The "Call to Action" section provides concrete, sensible next steps for researchers, industry, and the community, such as developing audio-native benchmarks and designing semantic audio tokens.

Weaknesses:
Empirical Evidence: The informal user survey (N=200) used to support the "habit inertia" argument is limited in scope. The authors admit it is not statistically representative, which weakens the behavioral claims central to their premise.

Cost Minimization: While the paper addresses compute cost in the appendix (estimating a 10-100x ratio), the main text might gloss over the sheer infrastructural hurdle of replacing highly efficient discrete text tokens with much denser audio tokens for massive pretraining runs. This is not insignificant.

**Support:**

2

---

> ### Author Rebuttal · Authors · 2026-03-29
>
> We thank the reviewer for the positive assessment and for precisely identifying where the argument can be made more rigorous.
>
> **Q1 (compute cost asymmetry):** The compute gap is real and we do not claim it will close soon. Audio at semantic token rates (~50-75 tokens/sec after hierarchical tokenization) is still 3-5x denser than text per unit of semantic content. At trillion-parameter scale this is a significant cost multiplier.
>
> But our argument is not that speech-native pretraining is cheaper - it is that the current cost asymmetry does not justify architectural neglect. The field invested heavily in scaling text-only models before the economics were favorable, because the scientific value was clear. We argue the same logic applies to speech-native architectures: the community should be exploring them now - at smaller scales, with efficient architectures, on targeted tasks - so that when compression and hardware close the gap futher the architectural foundations are ready.
>
> Treating cost as a permanent blocker rather than an active research target would be repeating the mistake of optimizing for today's data distribution while tomorrow's shifts underneath.
>
> **Q2 (survey demographics):** The survey was conducted as an informal online poll and carries self-selection bias toward technology-literate, younger respondents. This is precisely why the paper presents it as illustrative evidence of barrier patterns (footnote 1, page 6), not as a representative population study. The barrier ranking (speed > quality > single-turn > social discomfort > privacy) and the sharp home-vs-public usage gap are informative signal even under self-selection, but we will add an explicit limitations paragraph making the scope and caveats clearer.
>
> **Q3 (privacy):** Privacy is not a side issue we can hand-wave past. It is a first-order architectural constraint on speech-native systems. Any credible research agenda must include: consent-aware data pipelines distinguishing opted-in content from ambient capture, on-device processing for sensitive audio, federated learning adapted for speech, and differential privacy for continuous audio signals. The paper mentions these in Section 6 but we will elevate privacy from a bullet point to a core constraint. The GPT-4o system card already treats speaker identification and voice generation as model-specific safety risks - the research community should formalize this rather than treating it as a deployment afterthought.
>
> **Revision:** Strengthen privacy treatment in Section 6, add survey limitations paragraph in 4.1.

---

> > ### Author Rebuttal · Reviewer_8HwM · 2026-04-04
> >
> > Thanks to the authors for answering my questions, with the promised changes, I feel that my concerns are addressed.

---

### Official Review · Reviewer_U7QX · 2026-03-15

**Significance:** 4
**Argument Clarity:** 3
**Rating:** 5
**Confidence:** 5

**Questions:**

See weaknesses.

**Alternative Views Section:**

Yes

**Compliance With Llm Reviewing Policy A Conservative:**

Affirmed.

**Discussion Potential:**

4

**Final Justification:**

The interface-to-data feedback loop argument is the strongest part of this paper and I think it holds up well. The rebuttal's proposed revisions for W3 (carving out where text wins) and W4 (multilingual paragraph) should help. I still wish W2 (survey demographics) had been addressed since the barrier numbers lose weight without knowing who was surveyed, but it's not a dealbreaker. Keeping my score at 5.

**Paper Summary:**

The paper argues the ML community should prioritize speech-native foundation models because the data ecosystem is shifting toward speech-first. Four pillars: the interface-to-data feedback loop, speech tech being ready (Whisper, HuBERT, etc.), text-centric asymmetry in current multimodal models (AudioPaLM uses PaLM-2, SpeechGPT uses LLaMA, etc.), and habit inertia as the main barrier (backed by N=200 survey).

**Position:**

Yes

**Position In Title:**

Yes

**Related Work:**

4

**Strengths And Weaknesses:**

Strengths
1. I really like the "interface shapes expression shapes data shapes models" feedback loop in Section 1.1. As someone who works on audio, I've felt this asymmetry for a while but never seen it articulated this cleanly. It turns "speech is underserved" into a structural argument about how the whole field allocates attention.
2. I appreciate that the authors don't hide the hard parts. The compute cost analysis in Appendix A is upfront that audio is 10-100x more expensive, and the latency budget in Figure 6 (1200ms total vs 200ms human norm) is the kind of honest number that makes me trust the rest of the paper more.
3. The user survey (N=200) is a nice touch. Most position papers don't bother with any empirical grounding, so having actual numbers on barriers gives me something to point at.
4. I think the literature coverage on speech/audio is solid. The authors clearly know this space. They correctly nail the text-pretrained backbone problem across AudioPaLM, SpeechGPT, Qwen-Audio, and the GPT-4o/Gemini Live discussion is well-placed.

Weaknesses
1. My biggest issue is that the paper stays vague on what speech-native architectures would actually look like. GPT-4o and Gemini Live are cited but they're closed systems, no published details. If you're calling for a paradigm shift, I think you need to at least sketch what the new paradigm is, even speculatively.
2. I'm a bit worried about the survey methodology. N=200 is ok but there's no info on recruitment or demographics. If these are mostly tech-savvy early adopters, the barriers might look very different for the general population.
3. I felt the response to "text has genuine advantages" (Section 5.3) was too dismissive. Text really is better for search, editing, formal reasoning. I think the paper would be stronger if it carved out where text-first is still the right call, rather than framing everything as speech-first.
4. The whole thing is very English-centric. We know that ASR accuracy drops a lot for many languages and accents, and the case for speech-native gets both more urgent and harder to make there. Even a paragraph on this would help.
5. I get that it's a position paper, but even a small comparison between a speech-native and text-pretrained system on one downstream task would've made this much more convincing.

**Support:**

3

---

> ### Author Rebuttal · Authors · 2026-03-29
>
> We thank the reviewer for the generous and careful reading.
>
> **W1 (architecture vagueness):** This is the most common request across all three reviews and we take it seriously. However, we want to defend the paper's scope: this is a position paper arguing *why* the field should pursue speech-native architectures, not a systems paper proposing *which* architecture to build.
>
> That said, we can be more concrete about the research directions: (1) Audio-first pretraining - models trained on audio from the start rather than text-pretrained models adapted to audio. (2) Semantic audio tokenization - moving beyond phonetic clustering (HuBERT, wav2vec) toward representations preserving meaning and paralinguistic structure hierarchically. (3) Evaluation without text intermediation - benchmarks measuring reasoning over audio directly, not through ASR-then-NLU pipelines.
>
> These are research questions, not solved problems. The paper's job is to argue they deserve priority.
>
> **W3 (text advantages dismissed):** We partially agree. Section 5.3 can be sharper in carving out where text remains preferable: search/retrieval, non-linear editing, silent interaction, formal documentation, and accessibility for deaf users. But acknowledging text's advantages is not the same as accepting text as the permanent default. Both can be true - text is better for some tasks and speech deserves first-class architectural treatment because it carries signal that text discards.
>
> **W4 (English-centric):** Acknowledged. ASR accuracy drops significantly for low-resource languages and non-standard accents, and the speech-native argument is both more urgent (these communities are often speech-first in daily life) and harder to realize (less data, less research). We will add a paragraph stating multilingual robustness and accent equity are core research blockers, not optional future work.
>
> **W5 (downstream comparison):** We understand the desire but we do not have this experiment and we will not promise one we cannot deliver. This is a position paper; its contribution is the argument, not experimental evidence. We will note this as a high-priority open experiment the community should pursue.
>
> **Revision:** Expand Section 6.1 with three concrete research directions. Expand Section 5.3 with text-superiority carve-outs. Add multilingual/accent equity paragraph.

---

> > ### Author Rebuttal · Reviewer_U7QX · 2026-04-03
> >
> > Thanks for the response. W3 and W4 are addressed and the proposed revisions sound reasonable. However, W2 (survey methodology) was not addressed at all. I still don't know who the 200 respondents were or how they were recruited, which matters for interpreting the barrier numbers. Also, the three research directions for W1 (audio-first pretraining, semantic tokenization, text-free evaluation) are things most people in the audio community would already list. I was hoping for something more specific or surprising.

---

### Decision · Program_Chairs · 2026-04-30

**Decision:**

Accept (spotlight)

**Comment:**

The reviews are strongly positive overall, and the main concerns were either limited in scope or adequately addressed in rebuttal. Reviewers agreed that the paper has a clear position, is timely, and raises an important issue for the community. Its strongest contribution is not a new technical proposal, but a compelling reframing of current foundation model design: the paper argues that text dominance reflects historical interface conventions as much as intrinsic necessity, and that this bias now shapes data collection, model design, and evaluation.

The main weaknesses noted by reviewers were also clear: the survey evidence is limited, some claims around habit inertia were initially too strong, and the paper could be more concrete about what speech-native research directions look like. I think the authors responded well here. They clarified the intended role and limitations of the survey, narrowed some claims, and made the future research agenda more concrete without pretending to have solved the architecture question already. That is an appropriate response for a position paper.

Overall, it stakes out a clear view, engages competing perspectives, and is likely to stimulate useful discussion.